# Stable and expressive recurrent vision models

**Drew Linsley**[*], **Alekh K Ashok**[*], **Lakshmi N Govindarajan**[*], **Rex Liu, Thomas Serre**
Carney Institute for Brain Science
Department of Cognitive Linguistic & Psychological Sciences
Brown University
Providence, RI 02912
{drew_linsley,alekh_ashok,lakshmi_govindarajan,
rex_liu,thomas_serre}@brown.edu

## Abstract

Primate vision depends on recurrent processing for reliable perception [1–3]. A growing body of literature also suggests that recurrent connections improve the learning efficiency and generalization of vision models on classic computer vision challenges. Why then, are current large-scale challenges dominated by feedforward networks? We posit that the effectiveness of recurrent vision models is bottlenecked by the standard algorithm used for training them, "back-propagation through time" (BPTT), which has $\mathcal{O}(N)$ memory-complexity for training an $N$ step model. Thus, recurrent vision model design is bounded by memory constraints, forcing a choice between rivaling the enormous capacity of leading feedforward models *or* trying to compensate for this deficit through granular and complex dynamics. Here, we develop a new learning algorithm, "contractor recurrent back-propagation" (C-RBP), which alleviates these issues by achieving constant $\mathcal{O}(1)$ memory-complexity with steps of recurrent processing. We demonstrate that recurrent vision models trained with C-RBP can detect long-range spatial dependencies in a synthetic contour tracing task that BPTT-trained models cannot. We further show that recurrent vision models trained with C-RBP to solve the large-scale *Panoptic Segmentation* MS-COCO challenge outperform the leading feedforward approach, with fewer free parameters. C-RBP is a general-purpose learning algorithm for any application that can benefit from expansive recurrent dynamics. Code and data are available at https://github.com/c-rbp.

## 1   Introduction

Ullman (1984) famously theorized that humans reason about the visual world by composing sequences of elemental computations into "visual routines" [2]. It has been found that many of these visual routines, from perceptual grouping [4] to object categorization [5], depend on local and long-range recurrent circuits of the visual cortex [1, 3, 6]. Convolutional neural networks (CNNs) with recurrent connections – recurrent CNNs – also seem to learn visual routines that standard feedforward CNNs do not [7–14]. For example, consider the *Pathfinder* challenge in Fig. 1a, which asks observers to trace the contour extending from the white dot. Although *Pathfinder* is visually simple, clutter and variations in path shape make it difficult for feedforward CNNs to solve, even very deep residual networks [7,8]. By contrast, a one-layer recurrent CNN can learn to solve *Pathfinder* by incrementally grouping paths from one end to the other, reminiscent of Gestalt-like visual routines used by human observers [7, 8, 15]. Others have found that the visual routines learned by recurrent CNNs on small computer vision datasets lead to better sample efficiency and out-of-distribution generalization than feedforward CNNs [6, 16, 17]. There is also evidence that primate visual decisions and neural

---

[*]These authors contributed equally to this work.

responses elicited by natural images are best explained by recurrent CNNs [17–23]. Nevertheless, the great promise of recurrent CNNs has yet to translate into improvements on large-scale computer vision challenges like MS-COCO [24], which are dominated by feedforward CNNs.

A well known limitation of recurrent CNNs is a memory bottleneck imposed by the standard learning algorithm, "back-propagation through time" (BPTT; [25]). The memory requirement of BPTT-trained models scales linearly with steps of processing, since optimization involves propagating error through the full latent trajectory. This makes it difficult to develop recurrent CNNs that can rival the massive capacity of leading feedforward CNNs, which is critical for performance on challenges [26], while also simulating enough steps of processing to learn robust human-like visual routines.

**Contributions.** We develop a solution to the recurrent CNN memory bottleneck introduced by BPTT. Our work is inspired by recent successful efforts in memory-efficient approximations to BPTT for sequence modeling [27, 28]. Of particular interest is recurrent back-propagation (RBP), which exploits the stability of convergent dynamical systems to achieve constant memory complexity w.r.t. steps of processing [27, 29, 30]. This approach depends on models with stable dynamics that converge to a task-optimized steady state. However, we find that leading recurrent CNNs violate this assumption and "forget" task information as they approach steady state. While this pathology can be mitigated with hyperparameters that guarantee stability, these choices hurt model performance, or "expressivity". Thus, we make the observation that recurrent CNNs face a fundamental trade-off between stable dynamics and model expressivity that must be addressed before they can adopt efficient learning algorithms and compete on large-scale computer vision challenges.

- We derive a constraint for training recurrent CNNs to become both stable *and* expressive. We refer to this as the *Lipschitz-Constant Penalty* (LCP).
- We combine LCP with RBP to introduce "contractor-RBP" (C-RBP), a learning algorithm for recurrent CNNs with constant memory complexity w.r.t. steps of processing.
- Recurrent CNNs trained with C-RBP learn difficult versions of *Pathfinder* that BPTT-trained models cannot due to memory constraints, generalize better to out-of-distribution exemplars, and need a fraction of the parameters of BPTT-trained models to reach high performance.
- C-RBP alleviates the memory bottleneck faced by recurrent CNNs on large-scale computer vision challenges. Our C-RBP trained recurrent model outperforms the leading feedforward approach to the MS-COCO Panoptic Segmentation challenge with nearly 800K fewer parameters, and without exceeding the memory capacity of a standard NVIDIA Titan X GPU.

## 2   Background

We begin with a general formulation of the recurrent update step at $t \in \{1..N\}$ in an arbitrary layer of a recurrent CNN, which processes images with height $H$ and width $W$.

$$h_{t+1} = F(x, h_t, w_F). \tag{1}$$

This describes the evolution of the hidden state $h \in \mathbb{R}^{H \times W \times C}$ through the update function $F$ (a recurrent layer) with convolutional kernels $w_F \in \mathbb{R}^{S \times S \times C \times C}$, where $C$ is the number of feature channels and $S$ is the kernel size. Dynamics are also influenced by a constant drive $x \in \mathbb{R}^{H \times W \times C}$, which in typical settings is taken from a preceding convolutional layer. The final hidden state activity is either passed to the next layer in the model hierarchy, or fed into an output function to make a task prediction. The standard learning algorithm for optimizing parameters $w_F$ w.r.t. a loss is BPTT, an extension of back-propagation to recurrent networks. BPTT is implemented by replicating the dynamical system in Eq. 1 and accumulating its gradients over $N$ steps (SI Eq. 7). BPTT computes gradients by storing each $h_t$ in memory during the forward pass, which leads to a memory footprint that increases linearly with steps.

**Steady state dependent learning rules** There are alternatives to BPTT that derive better memory efficiency from strong constraints on model dynamics. One successful example is recurrent back-propagation (RBP), which optimizes parameters to achieve steady-state dynamics that are invariant to slight perturbations of input representations; a normative goal that echoes the classic Hopfield network [31]. When used with models that pass its test of stability, which

we detail below, RBP memory complexity is *constant* and does not scale with steps of processing (the precision of dynamics). RBP is especially effective when a system's steady states can be characterized by determining its Lyapunov function [27]. However, such analyses are generally difficult for non-convex optimizations, and not tractable for the complex loss landscapes of CNNs developed for large-scale computer vision challenges. RBP assumes that dynamics of the transition function $F(\cdot)$ eventually reach an equilibrium $h^*$ as $t \to \infty$ (Eq. 2).

$$h^* = F(x, h^*, w_F) \qquad (2) \qquad \qquad \Psi(w_F, h) = h - F(x, h, w_F) \qquad (3)$$

In other words, $h^*$ is unchanged by additional processing. We can construct a function $\Psi(\cdot)$ such that the equilibrium $h^*$ becomes its root (Eq. 3); i.e. $\Psi(w_F, h^*) = 0$. RBP leverages the observation that when differentiating Eq. 3, the gradient of steady state activity $h^*$ w.r.t. the parameters of a stable dynamical system $w_F$ can be directly computed with the Implicit Function Theorem [29, 30, 32, 33].

$$\frac{\partial h^*}{\partial w_F} = (I - J_{F,h^*})^{-1} \frac{\partial F(x, h^*, w_F)}{\partial w_F} \qquad (4)$$

Here, $J_{F,h^*}$ is the Jacobian matrix $\partial F(x, h^*, w_F)/\partial h^*$ (SI §3). In practice, the matrix $(I - J_{F,h^*})^{-1}$ is numerically approximated [27, 29]. RBP is designed for RNNs that pass its *constraint-qualifications* test for stability, where **(i)** $\Psi(\cdot)$ is continuously differentiable with respect to $w_F$ and **(ii)** $(I - J_{F,h^*})$ is invertible. When these conditions are satisfied, as is often the case with neural networks for sequence modeling, RBP is efficient to compute and rivals the performance of BPTT [27].

## 2.1 Do recurrent CNNs pass the constraint-qualifications test of RBP?

We turn to the *Pathfinder* challenge [7, 8] to test whether RBP can optimize recurrent CNN architectures devised for computer vision (see 1a and Fig. S2 for examples). *Pathfinder* is an ideal test bed for recurrent vision models because they can more efficiently solve it than feedforward models.

- *Pathfinder* tests the ability of models to detect long-range spatial dependencies between features – identifying the target contour and tracing it from one end to the other. Feedforward architectures (like ResNets) need a sufficient amount of depth to learn such dependencies, leading to an explosion of parameters and learning problems [7, 8]. In contrast, recurrent CNNs can broaden receptive fields over steps of processing without additional processing layers (SI §3), but doing so requires maintaining task information across dynamics.

- *Pathfinder* is parameterized for fine-grained control over task difficulty. By lengthening or shortening target contours and clutter, we can generate more or less difficult datasets (Fig. S2) while controlling other perceptual variables that could introduce spurious image/label correlations.

In summary, recurrent CNNs that can solve *Pathfinder* need to learn a dynamic visual routine for detecting long-range spatial dependencies in images. BPTT-trained recurrent CNNs can do this [7, 8]. Here we test whether RBP-trained models can do the same.

**Methods** We trained the leading recurrent CNN for *Pathfinder*, the horizontal gated recurrent unit (hGRU; a complete model description is in SI §3.1), to solve a version where the target contour is 14-dashes long (Fig. 1a). We modified *Pathfinder* from its origins as a classification task [12] to a segmentation task, which makes it easier to interpret model dynamics and translate our findings to the large-scale MS-COCO challenge examined in §3.2. The hGRU architecture consisted of **(i)** an input layer with 24 Gabor-initialized convolutional filters and one difference-of-Gaussians filter, followed by **(ii)** an hGRU with $15 \times 15$ horizontal kernels and 25 output channels, and finally **(iii)** a $1 \times 1$ convolutional "readout" that transformed the final hGRU hidden-state to a per-pixel prediction via batch normalization [34] and a $1 \times 1$ convolutional kernel. We began by testing four versions of the hGRU: one trained with BPTT for 6 steps, which was the most that could fit into the 12GB memory of the NVIDIA Titan X GPUs used for this experiment; and versions trained with RBP for 6, 20, and 30 steps. We also trained a fifth control model, a feedforward version of the 6 step BPTT-trained hGRU, where parameters were not shared across time ("feedforward CNN control"). The models were trained with Adam [35] and a learning rate of 3e-4 to minimize average per-pixel cross entropy on batches of 32 images. Training lasted 20 epochs on a dataset of 200,000 images. Performance was measured after every epoch as the mean intersection over union (*IoU*) on a held-out test set of 10,000 images. We report the maximum *IoU* of each model on this test set.

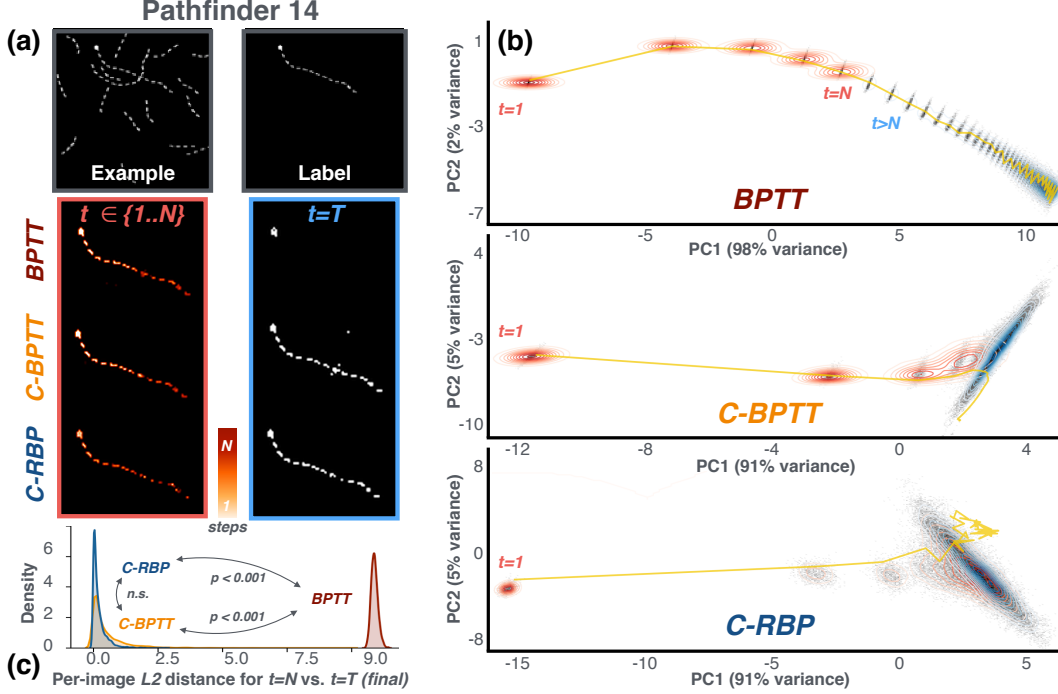

Figure 1: Recurrent CNNs trained with backpropagation through time (BPTT) have unstable dynamics and forget task information. This pathology is corrected by our *Lipschitz Coefficient Penalty* (LCP). **(a)** Incremental segmentations of horizontal gated unit (hGRU) models trained for $N$ recurrent steps on Pathfinder-14. Heatmaps in the left column show the models tracing the target contour until $t=N=6$ steps. The right column depicts model predictions after $t=T=40$ steps. Segmentations from the BPTT-trained hGRU degenerate but models trained with LCP did not. Learning algorithms of LCP-trained models are "contractor-BPTT" (C-BPTT) and "contractor-RBP" (C-RBP). **(b)** Visualization of horizontal gated unit (hGRU) state spaces by projecting hidden states onto each model's top-two eigenvectors. Grey dots are the 2D-histogram of projected hidden states, red contours are hidden state densities up to the task-optimized $N$ steps, and blue contours are hidden state densities following that step. Dynamics for an exemplar image are plotted in yellow. BPTT-trained model dynamics diverge when $t > N$, but models trained with LCP did not. **(c)** Two-sample KS-tests indicate that the distance in state space between $t = N$ and $t = T$ hidden states is significantly greater for an hGRU trained with BPTT than an hGRU trained with C-BPTT or C-RBP (n.s. = not significant).

**Results** The three hGRUs trained with RBP performed poorly on Pathfinder-14 (6 step: 0.50 *IoU*; 20 step: 0.71 *IoU*; 30 step: 0.70 *IoU*), far worse than a BPTT-trained hGRU (0.98 *IoU*). The RBP-hGRUs were also outperformed by the feedforward CNN control (0.78 *IoU*), although this control used 3 times more parameters than the hGRUs.

Why does an hGRU trained with RBP fail to learn *Pathfinder*? To address this question, we return to the constraint-qualifications test of RBP. The hGRU, like all models successfully optimized with gradient descent, satisfies condition (**i**) of the test: it is composed of a series of differentiable functions. This means that the problem likely arises from condition (**ii**), which requires the matrix $I - J_{F,h^*}$ to be invertible. Indeed, if $I - J_{F,h^*}$ is singular and not invertible, training dynamics will devolve into an unstable regime as we observed for RBP-hGRU training (Fig. S7). One way of guaranteeing an invertible $I - J_{F,h^*}$ is by forcing $F$ to be a contractive map [33]. When this constraint is in place, we can invoke Banach fixed point theorem to ensure convergence to a unique fixed point starting from any initial hidden state $h_0$. To elaborate, $F$ implements a contraction map and will converge onto a unique fixed point if the following holds: $\forall h_i, h_j, x$, where $i, j \in \mathbb{R}_+$, $\exists \lambda \in [0, 1)$ such that

$$\|F(h_i; x, w_F) - F(h_j; x, w_F)\|_2 \le \lambda \|h_i - h_j\|_2 \tag{5}$$

A straightforward way of forcing a contractive map is by building models with hyperparameters that are globally contractive (i.e., across the entire latent trajectory of dynamics), like squashing

non-linearities (*sigmoid* and *tanh* [36, 37]). However, these same hyperparameters are suboptimal for computer vision, where unbounded non-linearities (like *ReLU*) are used because of their control over vanishing gradients in deep hierarchical networks and improving function expressivity [38, 39].

In principle, recurrent CNNs like convolutional LSTMs, that use *tanh* non-linearities, are better suited for RBP optimization than an hGRU, which uses soft unbounded rectifications (*softplus*). Indeed, we found that a 20-step convolutional LSTM ("convLSTM"; architecture detailed in SI §3.2) trained with RBP on Pathfinder-14 performs slightly better (0.73 *IoU*) than the RBP-trained hGRU (0.71 *IoU*). At the same time, the RBP-trained convLSTM performs much worse than a 6-step BPTT-trained convLSTM (0.81 *IoU*) and a BPTT-trained hGRU (0.98 *IoU*). In other words, the convLSTM is more stable but less expressive than the hGRU. Furthermore, an hGRU with squashing non-linearities could not be trained reliably with RBP due to vanishing gradients. These findings raise the possibility that recurrent CNNs face a trade-off between "expressivity" needed for competing on large-scale computer vision challenges, and "stability" that is essential for using learning algorithms like RBP which rely on equilibrium dynamics.

**Expressivity vs. stability**  To better understand the trade-off faced by recurrent CNNs, we examined the stability of an *expressive* recurrent CNN: the BPTT-trained hGRU, which outperformed all other architectures on Pathfinder-14. The hGRU solves Pathfinder by incrementally grouping the target contour, making it possible to visualize the evolution of its segmentation over time. By passing the model's hidden states through its readout, we observed that task information vanishes from its hidden states after the task-optimized $N$-steps of processing (Fig. 1a; compare predictions at $t=N$ and $t=T$). These unstable dynamics were not specific to the BPTT-trained hGRU. We found similar results for a BPTT-trained convLSTM (Fig. S3), and when optimizing hGRUs with common alternatives to BPTT (Fig. S4). Next, we performed a state space analysis to measure model dynamics on all images in the Pathfinder-14 test set (method described in SI §3.3). The state space revealed a large divergence between hGRU hidden state activity at the task-optimized $t=6=N$ step vs. activity near steady state at $t=40=T$ (Fig. 1b). There was nearly as large of a difference between hidden states at $t=1$ and $t=N$ as there was between hidden states at $t=N$ and $t=T$.

# 3    Stabilizing expressive recurrent CNNs

While RNN stability is not necessary for all applications [40], it is critical for constant-memory alternatives to BPTT like RBP, which we hypothesized would improve recurrent CNN performance on large-scale vision challenges. Thus, we designed a "soft" architecture-agnostic constraint for learning local contraction maps, which balances model expressivity and stability over the course of training. Our goal was to derive a constraint to keep the largest singular value of $J_{F,h^*} < 1$, and force $F$ to be locally contractive at $h^*$ (SI §1.2; this contrasts with the global contraction across dynamics enforced by squashing non-linearities, which is problematic in computer vision as demonstrated in §2.1). However, the size of this Jacobian is quadratic in hidden state dimensions, making it too large to feasibly compute for recurrent CNNs. We overcome this limitation by introducing an approximation, our *Lipschitz Coefficient Penalty* (LCP), which constrains $(\mathbf{1} \cdot J_{F,h^*})_i < \lambda \, \forall i$, where $i$ is a column index.

$$\|(\mathbf{1} \cdot J_{F,h^*} - \lambda)^+\|_2 \tag{6}$$

Here, $(\cdot)^+$ denotes element-wise rectification and $\lambda \in [0, 1)$ is the hand-selected Lipschitz constant which bounds $\|J_{F,h^*}\|_2$ and hence the degree of contraction in $F$. The LCP (Eq. 6) can be combined with any task loss for model optimization, and circumvents the need to explicitly determine the Jacobian $J_{F,h^*}$. We show in SI §1.2 that the vector-Jacobian product used in the LCP, which is efficiently computed through *autograd* in deep learning libraries, serves as a approximate upper bound on the spectral radius of the $J_{F,h^*}$. Note that because minimizing the LCP implicitly involves the Hessian of $F$, it is best suited for recurrent CNNs that use activation functions with defined second derivatives (like *softplus* for the hGRU).

## 3.1    Stable and expressive recurrent CNNs with constant memory complexity

Returning to *Pathfinder*, we tested whether the LCP stabilizes the dynamics of models trained with RBP or BPTT. LCP consists of a single hyperparameter, $\lambda \in [0, 1)$, which mediates the degree of contraction that is imposed. As $\lambda \to 0^+$, the local contraction at a fixed point becomes stronger leading to increased stability and reduced expressiveness. In general we find that $\lambda$ is task dependent,

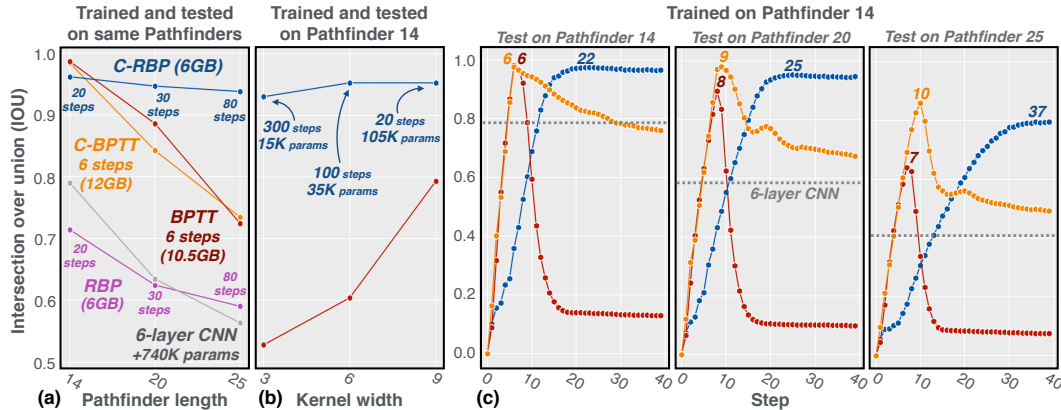

Figure 2: Enforcing contraction in recurrent CNNs improves their performance, parameter efficiency, and enables our constant-memory C-RBP learning algorithm. **(a)** hGRU models were trained and tested on different versions of *Pathfinder*. Only the version trained with C-RBP maintained high performance across the three datasets. **(b)** C-RBP models can rely on recurrent processing rather than spatially broad kernels to solve long-range spatial dependencies. BPTT-trained models cannot practically do this due to their linear memory complexity. **(c)** LCP improves the stability of hGRU dynamics and, as a result, the generalization of learned visual routines for contour integration. Models were trained on Pathfinder-14, and tested on all three Pathfinder datasets. hGRUs trained with C-RBP and C-BPTT generalized far better than a version trained with BPTT or a 6-layer CNN control. Numbers above each curve denote the max-performing step.

but on *Pathfinder*, models performed well over many settings. We set $\lambda = 0.9$ on the experiments reported here, and adopt the methods described in §2.

**Learning a task-optimal trade-off between stability and expressivity.** A BPTT-trained hGRU constrained with LCP performed as well on Pathfinder-14 as one trained with BPTT with no constraint (0.98 *IoU* for both). However, a state space analysis of this "contractor-BPTT" (C-BPTT) hGRU revealed stable contractor dynamics, unlike the BPTT-trained hGRU (Fig. 1b). We took this success as evidence that hGRUs trained with LCP satisfy the RBP *constraint qualifications* test. We validated this hypothesis by training a 20-step hGRU with RBP *and* LCP, which we refer to as "contractor-RBP" (C-RBP). This model performs nearly as well as both BPTT and C-BPTT trained hGRUs (0.95 *IoU*), despite using approximately half the memory of either. The C-RBP hGRU also converged to a steady state that maintained task information (Fig. 1b; $t = T$), and like C-BPTT, the distance between its task-optimized hidden state $t = N$ and steady state $t = T$ was smaller than the BPTT hGRU. Pairwise testing of these distances with 2-sample Kolmogorov–Smirnov (KS) tests revealed that the BPTT hGRU diverged from $t = N$ to $t = T$ significantly more ($\mu = 43.55$, $\sigma = 0.53$, $p < 0.001$) than either C-BPTT or C-RBP hGRUs ($\mu = 1.96$, $\sigma = 2.73$ and $\mu = 0.11$, $\sigma = 0.18$, respectively; Fig. 1c). We repeated these experiments for convLSTMs and found similar results, including C-RBP improving its performance on *Pathfinder* (Fig. S8). C-RBP therefore achieves our main goal of constant-memory training with performance on par with BPTT. However, in the following experiments, we demonstrate several qualities that make C-RBP preferable to BPTT.

**C-RBP can solve *Pathfinder* tasks that BPTT cannot.** We tested how recurrent CNNs performed on harder versions of *Pathfinder*, with 20- and 25-dash target paths, which forces them to learn longer-range spatial dependencies (Fig. S2). The linear memory complexity of BPTT caps the number of steps that can fit into memory, leading to poor performance on these datasets (Fig. 2a; BPTT and C-BPTT performance). Since C-RBP faces no such memory bottleneck, we were able to train 80-step C-RBP models to solve all versions of *Pathfinder*, achieving $> 0.90$ *IoU* on each dataset and vastly outperforming other models, including BPTT/RBP-trained hGRUs/convLSTMs, and the 6-layer CNN control model.

**C-RBP models can achieve better parameter efficiency through extensive recurrent processing.** *Pathfinder* is a test of contour integration, a visual routine for chaining arbitrarily small receptive

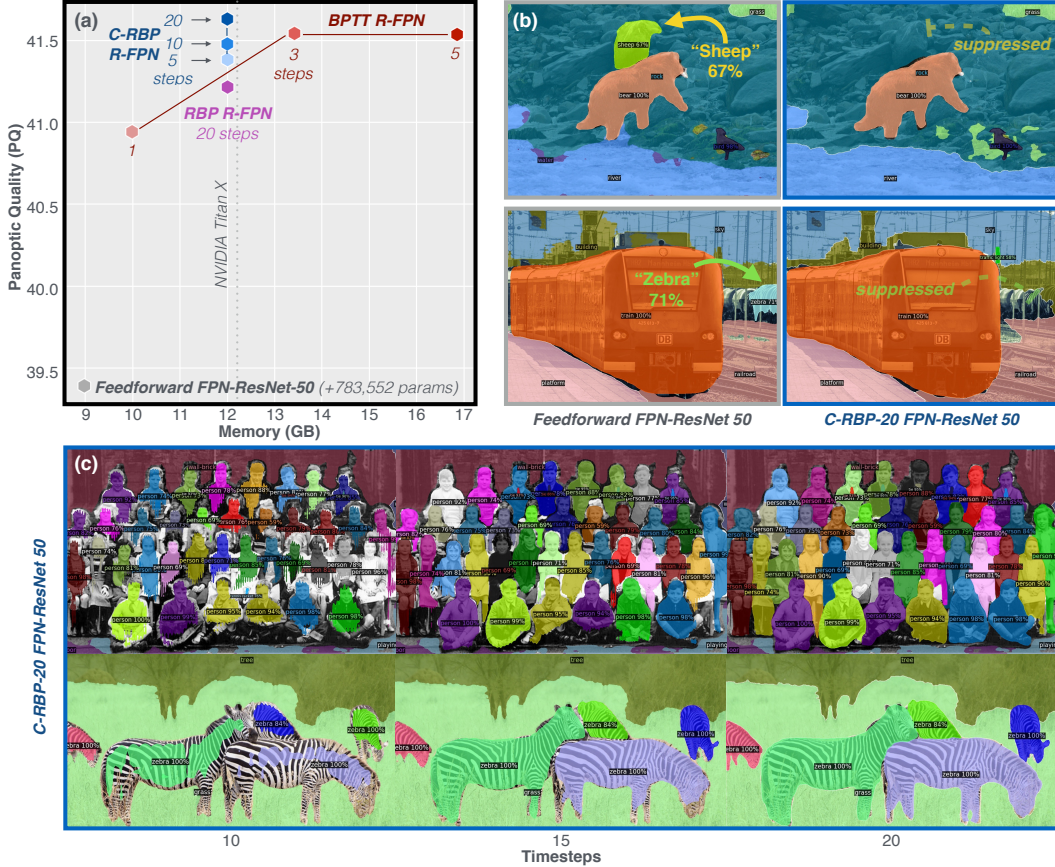

Figure 3: C-RBP trained recurrent vision models outperform the feedforward standard on MS-COCO Panoptic Segmentation despite using nearly 800K fewer parameters. **(a)** Performance of our recurrent FPN-ResNet 50 trained with C-RBP improves when trained with more steps of processing, despite remaining constant in its memory footprint. **(b)** Recurrent processing refines instance segmentations and controls false detections exhibited by the FPN-ResNet 50 (additional examples in SI). **(c)** Panoptic segmentation timecourses for an FPN-ResNet 50 trained with C-RBP for 20 steps.

fields together along the extent of the target contour. The instantaneous receptive field sizes of the recurrent CNNs used here is a function of convolutional kernel sizes. Thus, we hypothesized that models with small kernels could solve *Pathfinder* if they had sufficient steps of recurrent processing. Indeed, we found that hGRUs trained with C-RBP for 300 steps but only $3 \times 3$ kernels could solve Pathfinder-14 nearly as well as the baseline BPTT-trained hGRU with $15 \times 15$ kernels (Fig. 2b).

**Stable dynamics improve the out-of-distribution generalization of visual routines.** Visual routines such as contour integration derive long-range structure from repeated applications of a local grouping rule over steps of processing. This should lead to strong generalization on *Pathfinder*, even when testing out-of-distribution, or between datasets. For instance, increasing target contour length in *Pathfinder* affects the reaction time, but not accuracy, of human observers because they can solve the task through incremental grouping [8, 15]. If recurrent CNNs are learning similarly robust visual routines, a model trained on Pathfinder-14 should generalize to Pathfinder-20 and Pathfinder-25. To test this, we took hGRUs trained with BPTT, C-BPTT, or C-RBP on Pathfinder-14, and measured their performance on all versions of *Pathfinder* (Fig. 2c). Both C-BPTT and C-RBP trained hGRUs outperformed the BPTT-trained hGRU and feedforward control on each dataset, indicating that stable dynamics cause visual routines to generalize better. We found a similar improvement for C-RBP trained convLSTMs (Fig. S8). BPTT-alternatives were not as effective as C-BPTT or C-RBP (Fig. S8).

## 3.2 Microsoft COCO Panoptic Segmentation

Our main motivation in this paper is to understand – and resolve – the memory bottleneck faced by recurrent CNNs for large-scale computer vision challenges. To date, there have not been competitive recurrent solutions to the MS-COCO Panoptic Segmentation Challenge, a difficult task which involves recognizing the semantic category and/or object instance occupying every pixel in an image [41]. However, we will show that the stable dynamics and constant memory efficiency of C-RBP allows us to construct recurrent models that outperform the feedforward standards.

**Methods**   The leading approach to Panoptic Segmentation is an FPN Mask-RCNN [41] with a ResNet-50 or ResNet-101 layer backbone (refered to hereafter as FPN-ResNet-50 and FPN-ResNet-101). We developed recurrent extensions of these models, where we replaced the 4-layer Mask-RCNN head with an hGRU containing $3 \times 3$ kernels (SI §3.4). We call these models R-FPN-ResNet-50 and R-FPN-ResNet-101. MS-COCO Panoptic Segmentation involves two separate objectives: (**i**) instance segmentation of the discrete and enumerable "things" in an image, such as people and animals; and (**ii**) semantic segmentation of the remaining background elements in scenes, or "stuff" as it is known in the challenge. Models were trained to optimize both of these objectives with SGD+Momentum, a learning rate of 5e-2, and batches of 40 images across 24GB NVIDIA GTX GPUs (10 total). Model predictions on the COCO validation set were scored with Panoptic Quality (PQ), which is the product of metrics for semantic ($IoU$) and instance ($F_1$ score) segmentation [41]. Note that differences between models in PQ scores can be interpreted as a percent-improvement on instance and semantic segmentation for the total image. Recurrent models trained with LCP used $\lambda = 0.9$; training failed with higher values of this hyperparameter.

**Results**   After replicating benchmark performance of an FPN-ResNet-50 ($39.4PQ$, 9GB memory; `https://github.com/crbp`), we evaluated our R-FPN models (Fig. 3a). We first trained versions of the R-FPN-ResNet-50 with BPTT, which used nearly 17GB of memory for 5-steps of processing. BPTT performance increased with recurrence until it plateaued at 3-steps ($41.5PQ$ for both 3- and 5-step models). Next, we trained an R-FPN-ResNet-50 with RBP for 20 steps, and found that this model ($41.22PQ$, 12GB) performed better than the 1-step R-FPN-ResNet-50, but worse than a 2-step BPTT-trained model ($41.45PQ$). R-FPN-ResNet-50 models improved when they were trained with C-RBP. A 5-step model trained with C-RBP ($41.39$, 12GB) outperformed a 20-step model trained with RBP; a 10-step C-RBP model was similar ($41.48PQ$, 12GB) to the 3- and 5-step BPTT models; and a 20-step C-RBP model performed best ($41.63PQ$, 12GB). Importantly, each of these R-FPN-ResNet-50 models was more accurate than the feedforward FPN-ResNet-50 despite using 783,552 *fewer* parameters. We also found that a 20-step C-RBP R-FPN-ResNet-101 outperformed the feedforward FPN-ResNet-101 and BPTT-trained R-FPN-ResNet-101 models (Fig. S10).

There are two key qualitative improvements for the 20-step C-RBP R-FPN-ResNet-50 over the standard feedforward model. First, it suppressed false detections of the feedforward model, such as an over-reliance on texture for object recognition (Fig. 3b; other examples in Fig. S11-12). Second, and most surprisingly, the C-RBP model learned to solve the task by "flood-filling" *despite no explicit constraints to do so* (Fig. 3c). There is extensive work in the brain sciences suggesting that human and primate visual systems rely on a similar algorithm for object segmentation, classically referred to as "coloring" [2, 42]. We validated the extent to which the C-RBP R-FPN-ResNet-50 discovered a "flood-filling" segmentation routine by tessellating the ground-truth object masks in images, then applying a standard flood-filling algorithm to the resulting map. Both the C-RBP R-FPN-ResNet-50 and this idealized flood-filling model exhibited similar segmentation strategies (Fig. 4).

## 3.3 Related work

**Memory efficient learning algorithms**   BPTT's memory bottleneck has inspired many efficient alternatives. A popular example is truncated back-propagation through time (TBPTT), which improves memory efficiency with a shorter time horizon for credit assignment. There are also heuristics for overcoming the BPTT memory bottleneck by swapping memory between GPU and CPU during training [43, 44], or "gradient checkpointing", and recomputing a number of intermediate activities during the backwards pass [45–47]. Neural ODEs can train neural networks with continuous-time dynamics by optimizing through black box ODE solvers, but their computational demands and numerical issues result in poor performance on vision tasks ( [48, 48], see SI §4.1 for an expanded discussion). Deep equilibrium models (DEQ) are another constant-memory complexity learning

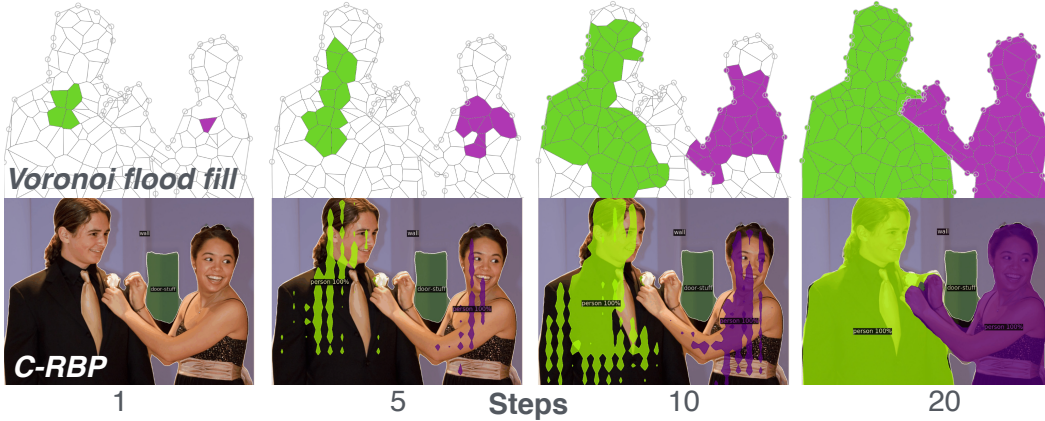

Figure 4: C-RBP-optimized models for Panoptic segmentation learn to implement a routine that resembles a flood-fill algorithm, despite no explicit supervision to do so. **Top** Flood-filling in a Voronoi-tesselation of object masks. **Bottom** The routine learned by a C-RBP R-FPN-ResNet-50.

algorithm with good performance on sequence modeling, rivaling the state of the art [28]. However, we found that training recurrent CNNs with DEQ is unstable, like with RBP (SI §4.1).

**Stable RNNs**   Recurrent model "stability" is typically analyzed w.r.t. training, where exploding or vanishing gradients are a byproduct of gradient descent [40, 49]. Models are stable if their Jacobians are upper-bounded by $\lambda$ (i.e., it is $\lambda$-contractive). One way to achieve this is with architecture-specific constraints on model parameters, which has worked well for generative adversarial networks (GANs) [50, 51] and sequence modeling LSTMs [40]. A simple alternative, is to directly penalize Jacobian approximations, as has been done for GANs [52–54]. However, these approximations have not been investigated for recurrent vision models or in the context of visual recognition.

## 4   Limitations and Future Directions

More work is needed to extend C-RBP to visual tasks other than the ones we analyzed here. For example, it is unclear whether C-RBP can extend to complex dynamical systems with limit cycles or strange attractors. It is possible that extensions are needed to transform C-RBP into a general method for balancing stability and expressivity when optimizing recurrent neural networks to explain arbitrary dynamical systems. We also did not examine complex recursive architectures where all recurrent layers are on the same dynamic "clock", such as in the $\gamma$-net of [17]. We leave the details of this application to future work, which we anticipate will be especially impactful for computational neuroscience applications, where such hierarchical feedback models can explain biological data that feedforward models cannot [8, 13, 17, 19, 22]. Another open problem is extending our formulation to spatiotemporal tasks, like action recognition, tracking, and closed-loop control. These tasks are defined by a time-varying feedforward drive, which presents novel optimization challenges.

## 5   Conclusion

There is compelling evidence from biological and artificial vision that the visual routines which support robust perception depend on recurrent processing [1–3]. Until now, the enormous cost of training recurrent CNNs with BPTT has made it difficult to test the hypothesis that these models can learn routines that will improve performance on large-scale computer vision challenges, which are dominated by high-capacity feedforward CNNs. C-RBP can alleviate this bottleneck, enabling constant-memory recurrent dynamics, more parameter efficient architectures, and systematic generalization to out-of-distribution datasets.

## Broader Impact

The development of artificial vision systems that can rival the accuracy and robustness of biological vision is a broadly useful endeavor for scientific research. Any such advances in artificial vision inevitably can lead to unethical applications. Because our methods and code are open sourced, our work is similarly exposed. However, we anticipate that our contributions to brain science make this risk worthwhile, and that our work will have a net positive impact on the broader scientific community.

## Acknowledgments and Disclosure of Funding

We thank Michael Frank and Xaq Pitkow for insights which motivated this work; and Alexander Fengler, Remi Cadene, Mathieu Chalvidal, Andrea Alamia, and Amir Soltani for their feedback. Funding provided by ONR grant #N00014-19-1-2029 and the ANR-3IA Artificial and Natural Intelligence Toulouse Institute. Additional support from the Brown University Carney Institute for Brain Science, Initiative for Computation in Brain and Mind, and Center for Computation and Visualization (CCV).

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
