[Supplementary Material]

# Supplementary Information
# Stable and expressive recurrent vision models

**Drew Linsley**\*, **Alekh K Ashok**\*, **Lakshmi N Govindarajan**\*, **Rex Liu, Thomas Serre**
Carney Institute for Brain Science
Department of Cognitive Linguistic & Psychological Sciences
Brown University
Providence, RI 02912
{drew_linsley,alekh_ashok,lakshmi_govindarajan,
rex_liu,thomas_serre}@brown.edu

## 1 Extended background

### 1.1 Backpropagation through time (BPTT)

BPTT is the standard learning algorithm for optimizing recurrent parameters $w_F$ w.r.t. a loss $\mathcal{L}(\tilde{y}, y)$. It is implemented by replicating a dynamical system and accumulating its gradients over $N$ steps of processing (Eq. 1, $K = 0$). Given a recurrent function $F$ parameterized by $w_F$, which maintains a latent state $h_t$ for each time step $t$, BPTT is implemented by Eq. 1.

$$\frac{\partial \mathcal{L}}{\partial w_F} = \frac{\partial \mathcal{L}}{\partial \tilde{y}} \frac{\partial \tilde{y}}{\partial h_T} \sum_{k=K}^{k=T-1} \left( \prod_{i=T}^{i=T-k} J_F(h_i) \right) \frac{\partial F}{\partial w_F}(x, h_{T-k}, w_F). \tag{1}$$

Here, $J_F(h_i)$ is the Jacobian of $F$ at $h$ on step $i$. Note that this algorithm stores every $h_t$ in memory during the forward pass, causing a memory footprint that linearly increases with steps.

### 1.2 Lipschitz Coefficient Penalty

We designed the *Lipschitz Coefficient Penalty* (LCP) as a hyperparameter-agnostic regularization for forcing recurrent CNNs to learn contraction maps. As mentioned in the main text, LCP constrains the vector-Jacobian product $(\mathbf{1} \cdot J_{F,h^*})_i < 1 \ \forall i$, where $i$ is a column index.

$$\|(\mathbf{1} \cdot J_{F,h^*} - \lambda)^+\|_2. \tag{2}$$

Here, $(\cdot)^+$ denotes element-wise rectification and $\lambda \in [0, 1)$ controls the degree of contraction in $F$. To derive LCP, we begin from the first-order Taylor expansion of $F(h)$,

$$F(h) \approx F(\bar{h}) + J_{F,\bar{h}} \cdot (h - \bar{h}) + \cdots,$$

with which one can show:

$$\frac{\left\| F(h) - F(\bar{h}) \right\|_2}{\left\| h - \bar{h} \right\|_2} \approx \frac{\left\| J_{F,\bar{h}} \cdot (h - \bar{h}) \right\|_2}{\left\| h - \bar{h} \right\|_2}, \tag{3}$$

Recalling the necessary condition for $F$ being a contractive map,

$$\left\| F(h) - F(\bar{h}) \right\|_2 < \lambda \left\| h - \bar{h} \right\|_2, \tag{4}$$

we can observe that the right hand side of Eq. 3 must be less than or equal to $\lambda \in [0, 1)$ for any $h$ sufficiently close to $\bar{h}$. Thus, $F(\cdot)$ will be $\lambda$-contractive *at least in the neighbourhood* of $\bar{h}$ if the LHS

is forced to be less than $\lambda$. We accomplish this goal by explicitly regularizing $\bar{h} = h^*$ over the course of training:

$$\|J_{F,h^*} \cdot \hat{v}\|_2 < 1, \tag{5}$$

for all unit vectors $\hat{v}$, which implies that the largest singular value of $J_{F,h^*}$ must be less than 1. This is equivalent to requiring $\|J_{F,h^*}\|_2 < 1$. Note that $\|h - h^*\|_2$ may not necessarily be small for all $h$'s sampled along our trajectories, and so the Taylor approximation and hence Eq. (3) may not hold. Nevertheless, in experiments on Pathfinder and Microsoft COCO our regularisation still yields reasonably stable convergence to fixed points.

Indeed, the matrix 2-norm is bounded from above and below by the 1-norm,

$$1/\sqrt{n}\,\|J_{F,h^*}\|_1 \le \|J_{F,h^*}\|_2 \le \sqrt{n}\,\|J_{F,h^*}\|_1\,, \tag{6}$$

where

$$\|J_{F,h^*}\|_1 = \max_i \sum_j \left|(J_{F,h^*})_{ij}\right|, \tag{7}$$

and $n$ is the dimensionality of the Jacobian matrix. So if we can regularize and force $\sqrt{n}\,\|J_{F,h^*}\|_1$ to be below 1, then we can ensure that $F(\cdot)$ will be contractive. However, computing Jacobians of large matrices requires an enormous memory load, and it is far more efficient to compute vector-Jacobian products instead. We shall instead approximate the 1-norm by taking

$$\max_i \left|\sum_j (J_{F,h^*})_{ij}\right| = \max_i (\mathbf{1} \cdot J_{F,h^*})_i\,, \tag{8}$$

where $\mathbf{1}$ denotes the row vector with all entries being 1. We note that in using this approximation, the right inequality in (6) ceases to be a strict upper bound, but we find that the approximation works well in practice due to large $n$.

We regularise model training with this approximation by requiring that $(\mathbf{1} \cdot J_{F,h^*})_i < 1 \ \forall i$. This yields our *Lipschitz Coefficient Penalty* (LCP):

$$\|(\mathbf{1} \cdot J_{F,h^*} - \lambda)^+\|_2, \tag{9}$$

which can be added to any task loss. Here, $(\cdot)^+$ denotes element-wise rectification and $\lambda \in [0, 1)$ is a hand-selected constant controlling the bound on $\|J_{F,h^*}\|_2$ and hence the degree of contraction in $F$. See Fig. S1 for an example implementation of LCP in pytorch.

```python
import torch

def LCP(last_state, second_last_state, mu=0.9):
    """Implementing the Lipschitz Coefficient Penalty with autograd."""
    norm_1_vect = torch.ones_like(last_state)   # Compute vector-jacobian product
    norm_1_vect.requires_grad = False
    vj_prod = torch.autograd.grad(
        last_state,
        second_last_state,
        grad_outputs=[norm_1_vect],
        retain_graph=True,
        create_graph=True,
        allow_unused=True)[0]
    vj_penalty = (vj_prod - mu).clamp(0) ** 2   # Clamp at mu and square
    return vj_penalty.mean()   # Minimize the average
```

Figure S1: Pytorch 1.4 code for computing the LCP.

## 2 Recurrent Back-prop

We review the Recurrent Back-Prop (RBP) learning algorithm of [1, 2]. Given a transition function $F$, which is parameterized by $w_F$ and applied to the static drive $x$, hidden state $h$ over $t \in \{1..N\}$ steps of processing: $h_{t+1} = F(x, h_t, w_F)$. We define a model readout, $\tilde{y} = G(h_T, w_G)$, where $G$ is

a task-optimized readout parameterized by $w_G$. We also introduce a loss $\mathcal{L}$ which yields a distance between predicted and ground-truth outputs. By differentiating the loss with respect to the weights, we obtain the gradients:

$$\frac{\partial \mathcal{L}_\infty}{\partial w_G} = \frac{\partial \mathcal{L}_\infty}{\partial y_\infty} \frac{\partial G(x; h^*, w_G)}{\partial w_G}, \tag{10}$$

$$\frac{\partial \mathcal{L}_\infty}{\partial w_F} = \frac{\partial \mathcal{L}_\infty}{\partial y_\infty} \frac{\partial y_\infty}{\partial h^*} \frac{\partial h^*}{\partial w_F}. \tag{11}$$

To obtain an expression for $\partial h^*/\partial w_F$, we first introduce the auxiliary function

$$\Psi(w_F, h) = h - F(x, h, w_F), \tag{12}$$

where, at a fixed point, $\Psi(w_F, h^*) = 0$. Differentiating with respect to $w_F$, we obtain

$$\begin{aligned}
\frac{\partial \Psi(w_F, h^*)}{\partial w_F} &= \frac{\partial h^*}{\partial w_F} - \frac{dF(x, h^*, w_F)}{dw_F} \\
&= \left( I - \frac{\partial F(x, h^*, w_F)}{\partial h^*} \right) \frac{\partial h^*}{\partial w_F} - \frac{\partial F(x, h^*, w_F)}{\partial w_F} \\
&= 0.
\end{aligned}$$

Rearranging yields

$$\frac{\partial h^*}{\partial w_F} = (I - J_{F,h^*})^{-1} \frac{\partial F(x, h^*, w_F)}{\partial w_F}, \tag{13}$$

where $J_{F,h^*}$ is the Jacobian matrix $\partial F(x, h^*, w_F)/\partial h^*$. The implicit function theorem guarantees the existence and uniqueness of a function mapping $w_F$ to $h^*$ and hence of $\partial h^*/\partial w_F$ provided (i) $\Psi(w_F, h)$ is continuously differentiable and (ii) $(I - J_{F,h^*})$ is invertible. The main virtue of RBP is that the memory it requires to train the RNN is constant with respect to the granularity of dynamics (steps of processing).

## 3   Recurrent vision models

### 3.1   *Pathfinder*

**Horizontal Gated Recurrent Units** The hGRU is a recurrent CNN, which when placed on top of a conventional feedforward convolutional layer, implements long-range nonlinear interactions between the feedforward layer's units [3]. These interactions take place over "horizontal connections" – a concept from neuroscience, in which anatomical connections between nearby cortical neurons (in retinotopic space) are the substrate for complex recurrent processing. Tracing back to its origins as a neural cir-

Figure S2: In our variation of the *Pathfinder* challenge [3,4], we ask observers to segment the contour connected to the white dot. Recurrent CNNs can easily solve it by learning to incrementally "trace" the target from end to end. Target contours are made up of 14-, 20-, or 25-dashes.

cuit model, the hGRU distinguishes itself from other recurrent CNNs as having two distinct stages of processing with independent kernels in each. The first stage computes suppressive interactions, i.e., a unit at location $(x, y)$ inhibits activity in a unit at location $(x + n, y)$, where $n$ is a spatial offset between these units. The second stage computes facilitative interactions, i.e., a unit $(x, y)$ excites activity in a unit at location $(x, y + n)$. The hGRU is governed by the following equations:

Stage 1:

$$\mathbf{A}^S = U^S * \mathbf{H}[t-1] \qquad\qquad\text{\# Compute channel-wise selection}$$

$$\mathbf{G}^S = sigmoid(\mathbf{A}^S) \qquad\qquad\text{\# Compute suppression gate}$$

$$\mathbf{C}^S = BN(W^S * (\mathbf{H}[t-1] \odot \mathbf{G}^S)) \qquad\qquad\text{\# Compute suppression interactions}$$

$$\mathbf{S} = \left[\mathbf{Z} - \left[(\alpha\mathbf{H}[t-1] + \mu)\,\mathbf{C}^S\right]_+\right]_+, \qquad\qquad\text{\# Additive and multiplicative suppression of } \mathbf{Z}$$

Stage 2:

$$\mathbf{G}^F = sigmoid(U^F * \mathbf{S}) \qquad\qquad\text{\# Compute channel-wise recurrent updates}$$

$$\mathbf{C}^F = BN(W^F * \mathbf{S}) \qquad\qquad\text{\# Compute facilitation interactions}$$

$$\tilde{\mathbf{H}} = \left[\nu(\mathbf{C}^F + \mathbf{S}) + \omega(\mathbf{C}^F * \mathbf{S})\right]_+ \qquad\qquad\text{\# Additive and multiplicative facilitation of } \mathbf{S}$$

$$\mathbf{H}[t] = (1 - \mathbf{G}^F) \odot \mathbf{H}[t-1] + \mathbf{G}^F \odot \tilde{\mathbf{H}} \qquad\qquad\text{\# Update recurrent state}$$

$$\text{where } BN(\mathbf{R}; \delta, \nu) = \nu + \delta \odot \frac{\mathbf{R} - \widehat{\mathbb{E}}[\mathbf{R}]}{\sqrt{\widehat{\mathrm{Var}}[\mathbf{R}] + \eta}}.$$

Here, $\mathbf{H}, \mathbf{Z} \in \mathcal{R}^{X \times Y \times C}$ are the hidden state and static drive from a preceding convolutional layer, respectively, with height/width/channels $X, Y, C$. Suppressive interactions in Stage 1 are computed with $W^S \in \mathbb{R}^{E \times E \times C \times C}$, and faciliative interactions in Stage 2 are computed with $W^F \in \mathbb{R}^{E \times E \times C \times C}$, where $E$ is the spatial extent of the horizontal connection kernel. In most of our experiments we set $E = 15$, as in [3] (other kernel sizes were tested in our parameter efficiency analysis in Fig. 2b). The hGRU also contains gates to modulate input activity and interpolate the previous hidden state with the current step's state, $U^S, U^F \in \mathbb{R}^{1 \times 1 \times C \times C}$. Steps of processing are indexed by $t \in \{1..N\}$, and rectification using *softplus* pointwise nonlinearities is denoted by $[\cdot]_+$, which ensures non-negativity in each stage, and hence, guarantees on suppression vs. facilitation. Lastly, we use batch normalization in the module to control exploding/vanishing gradients [5]. This introduces two learned kernels, $\delta$, $\nu \in \mathbb{R}^{1 \times 1 \times C}$, which control the scale and bias of normalization over input feature maps $\mathbf{R}$, and are shared across steps of processing ($\eta$ is a small constant that protects divide-by-zero errors). As is standard in batch normalization, $\widehat{\mathbb{E}}$ and $\widehat{\mathrm{Var}}$ are estimated on-line during training.

**Convolution LSTM**  We use a standard implementation of convolutional LSTMs from [6]. These models used kernels with the same dimensions as those described above for the hGRU.

## 3.2 State space analysis

We analyzed the state space of recurrent models trained to solve *Pathfinder*. This classic technique from dynamical systems has shown promise for analyzing computations of task-optimized recurrent neural networks [7]. Our approach to visualizing model state spaces involved the following steps: (**i**) Extract model hidden states for steps $t \in \{1..T\}$ elicited by a Pathfinder-14 image. (**ii**) Reduce hidden state dimensionality with a global average pool across spatial dimensions, yielding $C$-dimensional vectors. (**iii**) Fit a PCA using the $t \in \{1..N\}$ task-optimized steps of processing. (**iv**) Project all $t \in \{1..T\}$ hidden states onto the extracted eigenvectors.

## 3.3 Panoptic Segmentation

As a proof-of-concept of C-RBP on large-scale computer vision challenges, we developed a straightforward recurrent extension to the leading feedforward approach to the MSCOCO Panoptic Segmentation challenge: the FPN-ResNet. This model uses a ResNet backbone (either 50- or 101-layers) pretrained on ImageNet, which passes its activities to a feature pyramid network (FPN; [8]). FPN activities are then sent to a linear readout for semantic segmentation (to identify the "stuff" in images) and a Mask-RCNN for instance segmentation (to identify and individuate the "things"). We replace the Mask-RCNN head, which consists of 4-layers of convolutions and linear rectifications, with a single hGRU module (for both our version and the standard, activities from this stage are next

upsampled, rectified, and linearly transformed into predictions). The hGRU that we used is slightly different than the one for the *Pathfinder* challenge above. Batch normalization was replaced with group normalization [9], which is standard for Panoptic segmentation. We also used a modification of the input gate, following [10], which was found to improve performance for natural image processing. Standard feedforward FPN-ResNets were approximately twice as fast to train as our 20-step C-RBP R-FPN-ResNets. Models with ResNet-50 backbones took between one and two days to train, whereas models with ResNet-101 backbones took two and four days to train.

## 4 Extended discussion

### 4.1 Related work

**Recurrent vision models**    There are many successful applications of recurrent CNNS in computer vision, including object recognition, segmentation, and super-resolution tasks [6, 11–16]. These models often augment popular feedforward CNN architectures with local (within a layer) and/or long-range (between-layer) recurrent connections.

Others have found that augmenting recurrent CNNs with connectivity patterns or objective functions that are inspired by the anatomy or the physiology of the visual cortex can improve performance in visual reasoning, prediction, and recognition of occluded objects [6, 14–20].

**Lipschitz constraints for stable *training***    There are many examples of using constraints on Lipschitz continuity to stabilize deep network training. This is especially popular for generative adversarial networks (GANs), where stability is enhanced by constraining the spectral norm of the weights [21, 22], or Jacobians of each layer of the discriminator [23], or through Monte Carlo estimation of the discriminator's Jacobian [24, 25]. Penalizing the spectral norm of Jacobians can also yield better Auto Encoders [26], and adversarial robustness in CNNs [23]. In contrast to prior works on penalizing network Jacobians, in the current work we describe (i) an application to recurrent vision models, which (ii) enforces contractions only locally around equilibrium points (rather than globally across a hierarchy), which is a weaker constraint on model expressivity that still supports our key goal of stability during inference.

RNNs are notoriously challenging to train [27], and the classic solution is to constrain Lipschitz continuity by introducing learnable gates [28, 29]. It should be emphasized that our models also take advantage of gates, and while these control vanishing and exploding gradients to stabilize training, they are not sufficient to yield contractive mappings. More generally, it has been found that stability is critical to train RNNs that can solve sequence modeling tasks [30], but that stability is less critical for BPTT [29]. Other recent approaches induce stability via other architectural constraints, like weight orthogonalization via SVD [31].

**Deep networks as ODEs**    Neural ODEs exploit the observation that the residual networks can be treated as a discrete-time ODE. By using black-box ODE solvers in the forward and backward passes of the network, this discretization can be taken towards zero [32]. These models are trained with back-propagation through a latent trajectory derived from an adjoint system, giving them constant memory efficiency w.r.t. the granularity of dynamics, and they have shown promise in modeling continuous-time data and normalizing flows. However, Neural ODEs face several issues for computer vision applications. (*i*) Neural ODEs are difficult to optimize because input-output mappings become arbitrarily complex over the course of training. (*ii*) The adjoint method is slow to compute. (*iii*) Neural ODEs require feature engineering to fit certain classes of non-linear data manifolds [33], and (*iv*) they (along with the recent Augmented Neural ODEs [33]) do not compare favorably to standard feedforward models on simple computer vision benchmarks like CIFAR.

**Deep Equilibrium Models**    A recent extension to RBP includes a Neumann-series computation to approximate the inverse of a dynamical system's Jacobian [34]. Separately, deep equilibrium models (DEQ) use root-finding algorithms to exploit an identical formulation as in Eq. 3 to compute the steady state $h^*$ [35]. Both RBP and DEQ algorithms are effective for sequence modeling and meta-learning tasks, but have yet to be extended to vision. We attempted to train our hGRU models on Pathfinder 14 with DEQ, but it performed as poorly as the RBP-trained model, while using more GPU memory and taking longer to train.

Figure S3: Convolutional LSTMs trained with (BPTT) exhibit unstable dynamics, like the BPTT-trained hGRUs examined in the main text. Once again, LCP corrects this pathology. **(a)** Visualization of convLSTM and hGRU state spaces following the state space method described in Section . Here, the BPTT-LSTM was trained for 6 steps, the C-RBP LSTM for 60 steps, and the C-RBP hGRU for 40 steps. Grey dots are the 2D-histogram of projected hidden states, red contours are hidden state densities up to the task-optimized $N$ steps, and blue contours are hidden state densities beyond that point ($t > N$). Exemplar dynamics for a single image are plotted in yellow. While dynamics of the BPTT trained model diverge when $t > N$, models trained with LCP did not. **(b)** Model dynamics are reflected in their performance on Pathfinder-14 at $t = N$ and $t = T$ steps of processing. **(c)** Two-sample KS-tests indicate that the distance in state space between $t = N$ and $t = T$ hidden states is significantly greater for the BPTT-trained convLSTM than for either of the models trained with C-RBP (n.s. = not significant).

Figure S4: Additional state space analyses showed that alternatives to BPTT do not resolve the unstable dynamics we observed for recurrent CNNs. Here, BPTT per-step supervision refers to a model which was optimized with a loss evaluated on each of its 6 steps of processing. T-BPTT refers to a model trained with truncated backprop, for which gradients were accumulated over 3 steps of its 6 steps of processing. **(a,b,c)** These BPTT alternatives train models with unstable dynamics, which forgot task information after the optimized $t = N$ steps of processing. The distances between $t = N$ and $t = T$ hidden states are significantly greater for hGRUs trained with these algorithms than for an hGRU trained with C-RBP (n.s. = not significant).

Figure S5: Exemplars from the **(a)** *Pathfinder* challenge, along with **(b)** model performance on each of these datasets, and **(c)** predicted contours from hGRUs trained with the different algorithms.

Figure S6: Performance of hGRUs during training on *Pathfinder* challenge datasets. The RBP-trained model struggles to fit any dataset, unlike the models trained with BPTT, C-BPTT, or C-RBP.

Figure S7: The value of our LCP (computed with Eq. 6 in the main text) over the course of training for models that minimize it (C-RBP, C-BPTT) and models that do not (RBP, BPTT). In other words, the magnitude of this correlates with the stability/instability of model dynamics.

Figure S8: Generalization performance for hGRUs and convLSTMs trained with BPTT and alternatives to BPTT. Models were trained on Pathfinder 14 and tested on Pathfinder 14/20/25. For reference, performance of the hGRU trained with C-RBP is plotted in both rows. BPTT per-step loss means that a loss was computed on each of the 6 steps of hGRU training, and weights were optimized with BPTT. In contrast, a loss was only calculated on the final step for BPTT. TBPTT is truncated backprop through time, where gradients were computed over 3 steps of the 6 step dynamics. The LSTM trained with BPTT was trained for 6 steps, whereas the LSTM trained with C-RBP was trained for 60.

Figure S9: Recurrent model performance on MSCOCO Panoptic Segmentation. Performance was computed for each of 30 steps of processing for models trained with BPTT and C-RBP. The C-RBP models achieve better – and more stable – performance.

Figure S10: Our recurrent Panoptic Segmentation models also outperform the feedforward standard when both are given ResNet-101 backbones and the 3× training schedule (see `https://bit.ly/dtcon` for details on this training routine). The C-RBP model, trained for 20 steps, outperforms any other tested version of the model.

**Feedforward FPN-ResNet 50**      **C-RBP Recurrent FPN-ResNet 50**

Figure S11: Panoptic predictions from the feedforward ResNet-50 FPN Mask-RCNN (left) and our recurrent version of the model trained with C-RBP (right).

Figure S12: Panoptic predictions from the feedforward ResNet-50 FPN Mask-RCNN (left) and our recurrent version of the model trained with C-RBP (right).

## Footnotes

[1]These authors contributed equally to this work.