[Reviews · NeurIPS 2020]

Review 1

Summary and Contributions: In this paper, the authors present a method designed to overcome the memory bottleneck issues associated with backpropagation-through-time training of RNNS. The method builds on existing recurrent backpropagation (RBP) methods that can achieve O(1) memory complexithy (in contrast to O(n) for BPTT)), but adds in a Lipschitz-based penalty term that stabilizes the dynamics of the network, allowing a good tradeoff in expressivity-stability of the resulting models. Compelling results are shown for both a synthetic data problem (Pathfinder) and a more challenging real world-like application (MS-COCO panoptic segmentation).

Strengths: I found this paper to be compelling. Scalability issues with RNNs (particularly recurrent CNNs) are a huge issue limiting their broad application. In particular, the O(n) scaling in the number of timepoints considered limits the depth of networks that can be considered. The manuscript does a good job of explaining the limitations of the RBP method, which apply differentially to different problems, and demonstrating why the added penalty can at least give a model builder a knob with which to set an expressivity/stability tradeoff. The results presented with the Pathfinder data set are interesting, but I was also pleased that the authors showed good results with another difficult challenge problem. Given the promise of adding recurrence to CNNs in general, this new method appears to be promising advance.

Weaknesses: The addition of an additional hyperparameter always represents an additional complexity to model building, though in this case arguably not a particularly damning fault. I wonder if there are strategies for automatically modulating the contraction, akin to strategies for automatically adjusting learning rates over training. The choice of segmentation as a target task makes sense, given the intuitive advantage that recurrent connections would tend to provide, but it would have been nice to have also looked at other "less obvious" applications where a convolutional RNN may have been useful, such as standard classification. Similarly, it might have nice to explore other RNN types (including even cases where RBP is expected to perform well), to explore the generality of this approach outside of recurrent CNNs. However, I do not feel that these concerns are gating in any way... just nice-to-haves. minor: figure 4b: the examples are a little bit small to see clearly, but manageable minor: typo: "analysi,s" on page 6:218

Correctness: I believe the claims and methodology are correct.

Clarity: I found the writing to be clear and straightforward. The manuscript will benefit from some careful proofreading for minor typos.

Relation to Prior Work: Related work is clearly discussed within the page constraints afforded.

Reproducibility: No

Additional Feedback: I am satisfied with the author feedback, and I remain enthusiastic about this paper. I found the discussion between R4 and the authors regarding comparisons to methods on the leader board for the Panoptic Segmentation challenge to be helpful. Quite a few leader boards like this have "kitchen-sink" models dominating their top rank, which are in many cases difficult or even impossible to reproduce. While it would have been great if the authors could have reproduced the top method, I actually prefer looking at the simpler, more reproducible baseline.


Review 2

Summary and Contributions: This paper notes a major discrepancy between state-of-the-art computer vision algorithms and biological vision: the near-absence of recurrent computations in the former, and the abundance in the latter. The authors hypothesize that the dearth of recurrent models in computer vision is mainly due to the large memory cost of backpropagation through time (BPTT), which requires storing recurrent hidden state activations and therefore scales linearly with the number of recurrent time steps. To test this idea, they make two modifications to training recurrent networks: the Recurrent Back-Prop (RBP) algorithm, which has a constant memory cost, replaces BPTT; and a Lipschitz Coefficient Penalty (LCP) regularizes the Jacobian of the hidden state transition weights, which encourages the hidden state mapping to be locally contractive around its steady state. The authors then show that combining RBP with an LCP (C-RBP) makes recurrent dynamics stable over many more time steps than BPTT-trained RNNs, yet expressive enough to outperform feedforward or equally memory-intensive recurrent models on two vision tasks: target contour segmentation in "Pathfinder" datasets and panoptic segmentaton on MS-COCO.

Strengths: This is a rare submission that successfully identifies a key problem, develops a compelling theoretical explanation of the problem, and convincingly addresses it in a challenging domain (not toy tasks.) The illustration that the C-RBP method works on the Pathfinder dataset is fully convincing, and although the results on MS-COCO are not as dramatic, they're still an improvement on the most widely used method (which is at or near SOTA) and should generate many new ideas of how to incorporate RNNs into large-scale computer vision algorithms. The theory section of the paper is easy to follow, and the supplementary text makes clear what approximations are used in implementing RBP and the LCP.

Weaknesses: I wouldn't call it a weakness, but the relatively small improvement from using C-RBP over RBP and BPTT on MS-COCO makes me wonder whether the authors are being too conservative in their modification of ResNet-FPN into a recurrent network. Only the Mask-RCNN layers have been replaced by an RNN, whereas in principle the entire ResNet backbone could be made into a shallower recurrent model, and interaction between feature levels in the FPN could as well. I don't think any new experiments are necessary for this submission's acceptance, but I'm eager to see what happens when these methods are fully applied to large-scale computer vision models. Edits after seeing author response and other reviews: 1. Adding a future directions section to the main text seems like a good use of the extra space allowed in revision. The authors might also comment on whether there are any additional issues in using more "fully" recurrent architectures, not just the ones here that replace a small part of the ResNet-FPN-Mask-RCNN with their hGRU. 2. Reviewer 4 pointed out that, according to leaderboards, there are much more successful algorithms for doing panoptic segmentation on COCO-stuff test-dev. I think this deserves mention in the text, but I largely agree with the authors' response that many (most?) of these methods are not given enough detail to reimplement and also are not tested rigorously enough in their report to know where improvements are coming from (core architecture, hyperparameters, etc.) I think the authors took the right approach here of making precise changes to a well-known, publicly available baseline architecture (which is widely viewed to be a landmark for segmentation tasks.) Hence my confidence that, even though the performance/efficiency improvement is pretty small, it's due to the authors' proposed techniques.

Correctness: The experiments are well-designed and interpreted fairly.

Clarity: The paper is very clearly written. Edits: Taken as a whole, I still think this paper is very clear. Maybe it would be worth moving a little more of the math/(pseudo)code from the Supplement to the main text to illustrate exactly where the LCP is applied in the authors' implementation. It's challenging for any author to balance (1) intuitive explanations of the key concepts, (2) rigorous description of their techniques, and (3) a clear description of how to use the techniques in practice. I'd say this paper has all of the above, but given the exciting possibilities for future work it's probably worth honing this balance even more.

Relation to Prior Work: This paper combines several pieces of prior work in an exciting and (it turns out) useful way. It's very clear what the new contributions are. Edits after reading Author Response and other reviews: 1. It was clear to me on reading that the authors were not claiming to have developed all the techniques they combined here; just that the combination, and its application to training large-scale computer vision tasks, was new and successful. But I do see how readers could be mistaken that the pieces themselves are new, so it's worth explicitly saying this somewhere in the main text (maybe in bold.)

Reproducibility: Yes

Additional Feedback: I like the illustration of flood-filling in the Author Response: it's a very compelling example of an "interpretable" visual routine that emerges, unintentionally, from the combination of a new architecture (recurrent mask prediction) and a loss function/task (pixel-wise classification/segmentation.) If there's space in the revision, it would be nice to add this to the Discussion and compare it to what's known about the neurophysiology of object individuation.


Review 3

Summary and Contributions: An efficient replacement for BPTT is proposed, that can be applied to large CNNs with recurrent connections and vision tasks that require a substantial number of processing steps.

Strengths: Training large recurrent neural networks is of increasing importance, both for enhancing performance and for understanding the role of recurrent connections in the brain. The idea to make the recurrent dynamics locally contractive is is very nice, and new -as far as I know. Also the approach to focus on fixed points of recurrent dynamics is of interest from several perspectives. The algorithm is tested on hard versions of the Pathfinder and COCO Panoptic Segmentation Challenge. It is shown that their new method can solve instances of Pathfinder that BPTT cannot.

Weaknesses: The paper fails to mention that there are several online approximations to BPTT which do not run into the memory problem that motivates this submission (and do not require convergence of the dynamics). It would also be interesting to see whether

Correctness: Claims and methods are correct, as far as I can tell.

Clarity: Yes

Relation to Prior Work: In principle yes. But one could make clearer which of the aspects of the proposed methods are completely new, and which are improvements of approaches that have already previously been proposed.

Reproducibility: Yes

Additional Feedback:


Review 4

Summary and Contributions: This paper analyzes how recurrent networks could be made such that memory requirements would be constant (independent of the number of time steps used for training) without losing on performance. The authors enhance recurrent backpropagation (RBP) with a constraint that keeps network dynamics locally contractive near equilibrium. Architectures trained using this new constraint demonstrate the ability to integrate very long contours, work with small kernels, and generalize to untrained versions of Pathfinder challenge. The same idea also works somewhat better on the Panoptic Segmentation Challenge when compared to a standard feedforward architecture.

Strengths: I really enjoyed the thorough analysis of BPTT vs RBP and how to fix RBP to outperform BPTT on Pathfinder. The analysis is careful, clever, and well-supported by their findings. The method that they propose, namely, RBP with Lipschitz Coefficient Penalty, is generic, simple (see Fig. S1) and easily applicable. Using this method, the authors show impressive performance gains on Pathfinder challenge (to the point where we can declare it completely solved, I think). On top of that, they also show how this method could in principle work on the Panoptic Segmentation Challenge. I'm not fully convinced by the latter (see "Weaknesses") but I think the strength of this paper lies in the novel way to constrain recurrent networks and thus of great interest to NeurIPS community.

Weaknesses: One weakness that the authors acknowledged in supplementary materials is that this approach is meant for constant inputs and it is unclear what would happen if dynamic inputs were used. This is not a critical weakness (one step at a time!) but I thought is was an important one to keep in mind as the whole strength of recurrent networks seems to be for dynamic tasks. A more important issue in my view is the lack of convincing value for "real" tasks. Pathfinder is an excellent toy challenge to analyze networks and figure out how to improve them, but it is not really very interesting. Panoptic Segmentation Challenge is much better but there the authors show only slightly better performance than a standard approach. Also, I looked up the current leaderboard of this challenge (https://cocodataset.org/#panoptic-leaderboard) and the proposed approach would only rank 15, far behind the leading entry with PQ .547. Of course, this is not a fair comparison and maybe the same approach would improve the state-of-the-art, but it is impossible to know without trying. However, I am not familiar with this challenge and I also do not have a sense on the scale of PQ values. Perhaps I am missing something or being unfair, so I would be willing to improve my score if the authors can argue why I am incorrect in being skeptical about their approach's utility for the Panoptic Segmentation Challenge.

Correctness: As far as I can tell, the claims and method are correct but I did not verify derivations in the supplementary materials.

Clarity: No. Only on my second reading of this paper I started understanding what the paper was about. The authors have done so much work that it is very hard to explain everything in mere eight pages. I think quite a few details could be moved to supplementary materials and more emphasis placed on key findings. For instance, describing results in figures rather than text and using the gargantuan abbreviations only when absolutely needed would already be a great improvement (e.g., Line 145; Lines 165-170; Lines 220-232; Lines 280-292). Specifically, I think I would find the following structure more clear: Just have a single figure with a FF CNN, (C-)BPTT / (C-RBP) hGRU / convLSTM, so that we can clearly see how each additional thing helps. On that figure I would have time vs integrated path length (or IoU if that's easier to define). I'd expect to see the length of the integrated path grow until N steps, then start decreasing for BPTT but not for C-BPTT and C-RBP. A couple of insets could illustrate visually how different models are performing at t=N and t=T. This way, you could combine Figures 1 and 2 into a single figure (moving Fig. 2a and 2c to the supplementary) and hopefully spend much less time explaining how model X was better that Y because all that is clear from the figure (those details could go to the supplementary). Then, the main text could say: - We'll show how current approaches are suboptimal for Pathfinder (quick intro to this challenge) - hGRU is great but has memory issues - RBP is a great idea but doesn't work in practice - We need a contracting F - see convLSTMs for an inspiration - Our solution is LCP And then show Fig. 3 to demonstrate that C-RBP is better than C-BPTT. Minor: - Line 63: "on large-scale computer vision" – sentence not finished? - Line 137 mentions that 6 time steps were chosen because that was the most that could fit into 12GB GPU memory; but in Line 141 we learn that 4 GPUs were used. Maybe Line 137 could already mention the 4 GPUs as I found it pretty confusing. - Line 218: "analysi,s" → "analysis" - Line 222: "time model nearly as well" – a word missing?

Relation to Prior Work: Yes

Reproducibility: Yes

Additional Feedback: I have read the rebuttal and other reviewers comments. Thank you for all clarifications. I remain skeptical about the utility of C-RBP for Panoptic Segmentation; in fact, my prediction is that it would not improve state-of-the-art models. Nonetheless, I think this is not critical. The critical part of this work is providing a scalable way to build recurrent networks that can integrate over multiple steps. I think this work deserves at least a spotlight to encourage others to experiment with this approach, thus I upgraded my score to 8. Also, the additional figure in the rebuttal is really interesting!

[Author Response · NeurIPS 2020]

We thank the reviewers for their detailed feedback, and especially appreciate the suggestions on future research directions for C-RBP. Our revision will reflect the reviewers' comments, which we believe greatly improve its clarity.

**Future directions R1, R2, R4:** Each of the reviewers offered suggestions for future directions for our C-RBP method. We included a section *Future directions and limitations* in the SI, but we will move this to the main text and expand it in the final version of the paper to include the reviewer's suggestions, including the following:

**R1** Extending C-RBP to other tasks, like object classification. In this domain, we are especially excited to compare the decision strategies of a C-RBP-trained object classification model to human or non-human primate observers.

**R1, 2, 4** Extending C-RBP to other types of RNNs and data. The reviewers mention that our approach might have an even greater impact on other types of RNNs (*e.g.* graph neural networks) and datasets. We are particularly excited about extensions to domains with dynamic inputs, such as RL and spatiotemporal classification.

**Typos, formatting, and extended broader impact R1, R3:** We will fix these issues, thanks for pointing them out.

**Reviewing other online-approximations to BPTT R3:** We discuss some heuristics for controlling RNN memory complexity in the main text, such as gradient checkpointing, but the reviewer is right to point out that there are many other methods that we did not have room to include. The reviewer's comment was cut off, so we would appreciate additional guidance. One thought is that we could discuss online-approximations to BPTT motivated by Neuroscience, such as (1), which introduces eligibility traces to approximate BPTT-derived gradients. Note that this approach hasn't been extended to large-scale computer vision challenges, which was the goal of our work.

**Details for reproducibility R1:** We included code and data for model training, dataset creation, and generating results/figures in the SI zip file. We apologize for any confusion on how to run the code. In our revision we will include a link to a GitHub repository containing these files. Please advise on other details you'd like us to include.

**Streamlining manuscript organization R4:** We will move Figure 1 and lengthy explanations of the *Pathfinder* challenge/results to the SI. However, we would like to keep Figure 2 because it is an important motivation for C-RBP: BPTT-optimized RNNs can clearly solve segmentation tasks, but might do so using a sub-optimal dynamic routine.

**An intuitive explanation of Panoptic Quality R4:** Panoptic Quality (PQ) is sensitive to object recognition and instance/semantic segmentation, making it a difficult challenge. The baseline FPN-ResNet-101 PQ (43.0) vs. the baseline FPN-ResNet-50 PQ (39.4) implies the former improves segmentation on $3.60\%$ of an image. In contrast, our R-FPN-ResNet-50 (41.6 PQ) outperforms a FPN-ResNet-50 by $2.23\%$ despite having $\sim$1M fewer parameters.

**Skeptical about the utility of C-RBP for Panoptic Segmentation R4:**

- Our goal was to understand the effect of C-RBP and recurrent processing on Panoptic Segmentation while controlling for the effects of hyperparameters and pre/post-processing routines used on the challenge. Few of the models on the Panoptic Segmentation leaderboard are published, and those that are rely on loss function engineering and heuristics like test-time augmentations, ensembling, etc. Thus, we stuck with the FPN-ResNet as a strong baseline for a difficult challenge and demonstrated that an R-FPN trained with C-RBP outperforms it.

- We agree with the reviewer's prediction that *"the same [C-RBP] approach would improve the state-of-the-art"*. Indeed we offer evidence that C-RBP is generally useful for Panoptic Segmentation. An R-FPN-ResNet-50 trained with C-RBP outperforms a standard FPN-ResNet-50 (Figure 4). We also find that an R-FPN-ResNet-101 outperforms a standard FPN-ResNet-101 (Fig S9). R-FPNs use $\sim$1M fewer parameters than their respective FPN baselines.

- In retrospect, we regret not placing more emphasis on the *visual strategy* learned by an R-FPN trained with C-RBP for solving Panoptic Segmentation, which serves as qualitative evidence of our approach's *"convincing value for 'real' tasks"*. The R-FPN learns to segment objects by "flood-filling" – without any supervision or constraints to do this. It first seeds objects at their center then recurrently fills them to their boundaries (Fig. 1). This strategy gives models flexibility when segmenting and resembles theory from cognitive science on the visual routines of human observers for object segmentation. In our revision we will link to a website with animations that clarify this point.

Figure 1: R-FPNs trained with C-RBP learn to segment objects by "flood-filling" without any instruction to do so.

# References

[1] Roth, C., Kanitscheider, I., Fiete, I.: Kernel rnn learning (kernl). In: ICLR. (2018)


[Meta-Review · NeurIPS 2020]

This paper addresses the limitations of BPTT by proposing a new method (C-RBP) with O(1) memory-complexity. The proposed approach is evaluated on Pathfinder, showing reasonably good results. Reviewers were unanimously positive, though there were some minor concerns about clarity and the baselines used. I found the authors' response compelling in response to both points, as did the reviewers, though I would strongly encourage the authors to take the clarity suggestions seriously, as I feel they will significantly improve the paper. I recommend this paper should be accepted as a spotlight.